# Identification of Early Esophageal Cancer by Semantic Segmentation

**DOI:** 10.3390/jpm12081204

**Published:** 2022-07-25

**Authors:** Yu-Jen Fang, Arvind Mukundan, Yu-Ming Tsao, Chien-Wei Huang, Hsiang-Chen Wang

**Affiliations:** 1Department of Internal Medicine, National Taiwan University Hospital, Yun-Lin Branch, No. 579, Sec. 2, Yunlin Rd., Dou-Liu 64041, Taiwan; toby851072@gmail.com; 2Department of Internal Medicine, National Taiwan University College of Medicine, No. 1 Jen Ai Rd. Sec. 1, Taipei 10051, Taiwan; 3Department of Mechanical Engineering, Advanced Institute of Manufacturing with High Tech Innovations (AIM-HI), Center for Innovative Research on Aging Society (CIRAS), National Chung Cheng University, 168, University Rd., Min Hsiung, Chiayi 62102, Taiwan; d09420003@ccu.edu.tw (A.M.); tony00013@gmail.com (Y.-M.T.); 4Hitspectra Intelligent Technology Co., Ltd., 4F., No. 2, Fuxing 4th Rd., Qianzhen Dist., Kaohsiung 80661, Taiwan; 5Department of Gastroenterology, Kaohsiung Armed Forces General Hospital, 2, Zhongzheng 1st. Rd., Lingya Dist., Kaohsiung 80284, Taiwan; 6Department of Nursing, Tajen University, 20, Weixin Rd., Yanpu Township, Pingtung 90741, Taiwan

**Keywords:** esophageal cancer, small data, semantic segmentation, encoder–decoder model, U-Net, ResNet150V2, white light imaging, narrowband imaging

## Abstract

Early detection of esophageal cancer has always been difficult, thereby reducing the overall five-year survival rate of patients. In this study, semantic segmentation was used to predict and label esophageal cancer in its early stages. U-Net was used as the basic artificial neural network along with Resnet to extract feature maps that will classify and predict the location of esophageal cancer. A total of 75 white-light images (WLI) and 90 narrow-band images (NBI) were used. These images were classified into three categories: normal, dysplasia, and squamous cell carcinoma. After labeling, the data were divided into a training set, verification set, and test set. The training set was approved by the encoder–decoder model to train the prediction model. Research results show that the average time of 111 ms is used to predict each image in the test set, and the evaluation method is calculated in pixel units. Sensitivity is measured based on the severity of the cancer. In addition, NBI has higher accuracy of 84.724% when compared with the 82.377% accuracy rate of WLI, thereby making it a suitable method to detect esophageal cancer using the algorithm developed in this study.

## 1. Introduction

Esophageal cancer (EC) is an extremely dangerous and minimally researched cancer [1,2,3,4]. It is the eighth leading cause of cancer-related mortality and the sixth most common cancer type [5]. By the end of this decade, at least one in a hundred men in European countries such as the United Kingdom and the Netherlands will have EC [6]. Two out of five patients with EC are likely to be detected in late stages; hence, less than 20% of the patients survive more than 3 years [7,8,9]. Nevertheless, if the disease is diagnosed in the early stages, then the five-year survival rate will reach more than 90%. On the contrary, the rate will decrease to less than 10% when the disease is detected in the later stages. Thus, early detection of EC is important for increasing the survival rate [10,11]. At present, endoscopists are unable to draw a conclusion from the endoscope images of the esophagus during the early stages of EC [12]. Therefore, the disease can go unnoticed in the earlier stages. A precise assessment of individualized treatment depends on the accuracy of the initial diagnosis [13]. The endoscopic images acquired by using generic mechanisms will be altered by tissue secretion or instrument specifications, which may directly or indirectly lead to misjudgments in diagnosis. Computer-aided diagnosis (CAD) algorithms can be categorized based on the nature of the endoscopic image, either by narrowband image (NBI) or by white light imaging (WLI). These techniques use the penetration attributes of light. WLI utilizes a broad range of visible light to characterize the mucosa. Conversely, for NBI, two filters are placed on top of the light source, in the middle wavelength range: blue (415 nm) and green (540 nm) [14]. The infiltration of the blue filter is smaller than long-wavelength light. Nonetheless, it corresponds to the absorption of hemoglobin, thereby enabling veins, capillaries, and other parts with a higher hemoglobin ratio to appear darker and generate an adequate amount of contrast to the enclosing mucosa that reflects light. The second wavelength (540 nm) light corresponds to the secondary hemoglobin absorption peak; thus, deeper mucosal and submucosal vessels are made evident by the 540 nm light and are displayed in cyan [15,16].

In the past few years, considerable research has been published based on deep learning models, CAD, and other methodologies for diagnosing EC, and most of this research has shown potential for application [17,18,19,20,21,22,23,24,25,26,27,28,29]. For example, one study was conducted by Shahidi et al. by using the WLI and NBI to precisely detect EC through artificial intelligence (AI) [30]. Another study was conducted by Yoshitaka et al. by employing convolutional neural networks (CNN) to ascertain the invasion depth of EC under WLI. Results showed that using AI can be efficient to a greater extent for detecting esophageal squamous cell carcinoma (ESCC) than endoscopists [31]. Hiromu et al. conducted a study similar to CAD studies for the identification of ESCC and found that all CAD studies had better sensitivity and accuracy [32]. Wang et al. established a single shot multibox detector by using a CNN for identifying EC. Another study conducted by de Groof et al. built a hybrid ResNet-UNet model CAD algorithm [33]. Size et al. also built a ResNet-Based FCN backbone network to recognize cancerous areas in the esophagus [34]. Barbeiro et al. (2019) used NBI to study gastrointestinal endoscopy and found that NBI can be used as an important auxiliary image for WLI, which can improve the detection of gastrointestinal lesions. Apart from CAD, many biosensors have been recently researched to detect different cancer types [35,36,37]. However, most of these biosensors have a high limit of detection, thereby leading to poor sensitivity in early-stage cancer detection. With the increase in the domestic air-pollution rate, the risk for EC is constantly increasing [38,39]. Despite the rapid development of AI in recent years, EC continues to have the lowest survival rate because its symptoms are difficult to detect. Most of the recent research results are displayed in the form of box selection, which is less accurate in predicting the disease compared with semantic segmentation.

Therefore, this study aims to use the concept of semantic segmentation and U-Net as the basic artificial neural network, and Resnet to extract feature maps that will classify and predict the location of EC in its early stages.

## 2. Materials and Methods

### 2.1. Image Pre-Processing

As shown in Figure 1, the data used in this experiment are endoscopic images, and all endoscopic images generally have black frames. These black frames are unhelpful data in the training context. Therefore, the black frame was cropped; thus, only the esophagus image was provided for training, and the size of the image was modified to 608 pixels × 608 pixels. Another screening was conducted to ensure that the heavily blurry image and the images with rainbow lens flare or bubble interference were removed. All these images must be removed because the data set was already small, and in this case, any small interference could significantly affect the training set. Afterward, the doctors marked the area of the images with the symptoms. Then, these data were organized into two types: WLI and NBI. The training/validation set accounted for 90% of the total data, and the test set accounted for 10% of the total data. The number of images for both categories was as follows:WLI: 67 train/validation sets plus eight test setsNBI: 81 training/validation sets plus nine test sets

After allocating the data, the markers in the image were divided into two categories. The 0th category was normal, and the 1st category was a combination of dysplasia and SCC. All the data were saved as a .npy file to increase the speed of reading the data. Data augmentation was performed to increase the amount of data using the function ImageDataGenerator from the Keras library. The rotation range was set to 60, within which the images were rotated randomly. The sheer range was also set to 0.5, which will randomly apply shearing transformations.

### 2.2. Network Architecture

In this research, ResNet152V2+U-Net was chosen as the network design. Typically, the contracting path conforms to the design of a conventional convolutional network. It is comprised of the repeated application of two 3 × 3 convolutions (unpadded convolutions), each followed by a rectified linear unit (ReLU) and a 2 × 2 maximum pooling operation with stride 2 for downsampling [40,41]. However, the algorithm developed in this study was built as U-Net while the contracting path was updated to Res-Net152V2. Four of the five convolutional blocks of ResNet matched the number of convolutional blocks in the original U-Net expansion path. The respective contraction and expansion paths were then joined. The contraction path used Res-Net152V2, whilst the expansion path utilized the original U-Net. This design emphasized the use of a more complex model to locate a superior feature map. Multiple testing on the contracting route using the original U-Net, VGG19, and ResNet50 resulted in a significant accuracy improvement. As the input, the block from the first layer of ResNet152V2 was used. The block transferred to the expansion route was made compatible by an iterative procedure. Conv1 was utilized for the second layer, and the result of Conv1 was transferred to the extension path for connection. The third, fourth, and fifth levels then used Conv2 x, Conv3 x, and Conv4 x, respectively, of which one may be used to link the extension path layer. Its model architectures diagram are shown in Appendix A.

## 3. Results

The results of this study are divided into two parts: WLI and NBI. A total of 67 images were included for the training/validation set in WLI, whereas 81 images were included for the training/validation set in NBI. During training, WLI and NBI were trained simultaneously. The first 25 epochs used a learning rate of 0.001, and the learning rate of the last 10 epochs was revised down to 0.0002. The batch size was set to 1, and the number of steps was the number of samples. For the loss function, categorical_crossentropy was used, and for accuracy, categorical_accuracy was used because this study involved a multi-classification problem. The training accuracy was almost stable, and it gradually increased in the first 25 epochs before training. The verification accuracy was also higher than the training accuracy except for a few unstable ones. The training accuracy of the next 10 epochs gradually increased, and the verification accuracy was stable. The overall training effect was also good, and the gradual increase of the training accuracy was controlled at a good number of epochs to avoid overfitting. The verification accuracy of the 25 epochs before NBI training was higher than the accuracy during training except for a few instabilities. The accuracy rate of the 10 epochs after training was constantly changing and gradually increasing, whereas the verification accuracy rate was constantly changing and gradually decreasing close to the training accuracy rate. Despite a potential trend of overfitting, the data have also been expanded; thus, several epochs must be considered to achieve the best training. Using the trained model for the test set, WLI and NBI have eight and nine prediction sets, respectively. Several evaluation criteria were used for the results. Sensitivity is also known as a true positive or recall rate, which indicates the probability of successfully detecting positive samples among all true positive samples. The second criterion is precision, which indicates the probability of successfully detecting positive samples among all predicted positive samples. The F1-score is a harmonic average function of precision and recall, which is a rough indicator for checking the performance of this model. A confusion matrix shows the prediction results of the test set, which are presented in a tabular form. For the final result of this study, the three abovementioned evaluation indicators can be calculated based on the results of the confusion matrix. Figure 2 shows a schematic diagram of the results of this research. Figure 2a,b show the input WLI images after cropping, whereas Figure 2c,d are the input NBI images after cropping the original image. Figure 2e–h are ground-truth images marked by doctors based on the corresponding input images, whereas Figure 2i–l show the prediction results of the corresponding graphs. The orange-marked area corresponds to the category SCC, whereas the purple-marked area corresponds to the category dysplasia. Semantic segmentation can intuitively locate the marked position through the diagram. However, as shown in Figure 2i, some extra areas have EC. However, in the endoscopic image, a small lens flare is detected (Figure 2a). The similarity between the ground-truth and predicted results can be observed from Figure 2k. As shown in Figure 2j, two categories were marked similarly, which outperformed the detection of an expert doctor. However, in the NBI part, the symptoms of SCC and dysplasia were almost perfectly marked. Therefore, the NBI images can detect EC better than WLI images.

The accuracy, sensitivity, and F1 score were marked based on the severity of the diseases into three stages: normal, dysplasia, and SCC. The results of NBI are shown in Table 1, whereas the results of WLI are shown in Table 2. The overall accuracy rate was 84.7245% in NBI. The precision of dysplasia and SCC categories was 85.67%, which was higher than the accuracy of the normal category. However, in WLI, the accuracy was reduced to 82.37%. Moreover, the precision rate of dysplasia and SCC categories was only 77.24%, which was much lower than the required medical standards. Nevertheless, the accuracy of the normal category was 85.89%. Based on these results, we can infer that the NBI is a suitable method to detect EC in its early stages using the algorithm developed in this study.

## 4. Discussion

In recent years, the research on AI, biosensors, and medical treatment has constantly developed new methods to increase the probability of recovery after illness and reduce the possibility of morbidity or serious illness. Esophageal cancer has a low survival rate. Endoscopy can help doctors to grasp the location of the disease for the first time, no longer the approximate location, and it can also help patients reduce the chance of suffering from the use of iodine dyes. Semantic segmentation can intuitively see the marked position through the map. In the method developed in this study, particularly the NBI, SCC, and dysplasia symptoms were marked with high accuracy. In addition, NBI images can detect EC in its early stages better than WLI images in dysplasia and SCC, with high accuracy. Concerning accuracy, the symptoms are similar to the pixel data of the general esophagus. However, the amount of data in the training/validation set is relatively small, and the amount of disease information is insufficient. Hence, some results were not accurate. Furthermore, the accuracy of the developed model can be increased drastically by increasing the number of NBI and WLI images used for the training set.

## 5. Conclusions

Semantic segmentation is a method of classifying each pixel, and the information obtained from every pixel is important. When this technique is used in medical imaging, various abnormal areas can be marked. However, the symptoms of EC are negligible. Hence, the number of training and testing data as well as the resolution will be important factors in detection and classification. If the color of a single pixel is not accurate enough, then the model will have a color difference error when predicting the pixel and reproducing ground truth. However, the acquisition of endoscopic data is difficult. In this study, semantic segmentation was used to predict and label EC. ResNet152V2, and U-Net were also used as the AI architecture. In addition, a total of 165 images have been used, 67 of which were used for training and verification, whereas eight images were used for prediction in WLI images. A total of 81 images were used for training and verification, whereas nine images were used for prediction in NBI images. The time taken to predict each image in the test set is only 111 ms. The results indicate that the NBI has higher accuracy of 84.724% when compared with the 82.377% accuracy rate of WLI, thereby making it a suitable method to detect EC using the algorithm developed in this study.

## Figures and Tables

**Figure 1 jpm-12-01204-f001:**
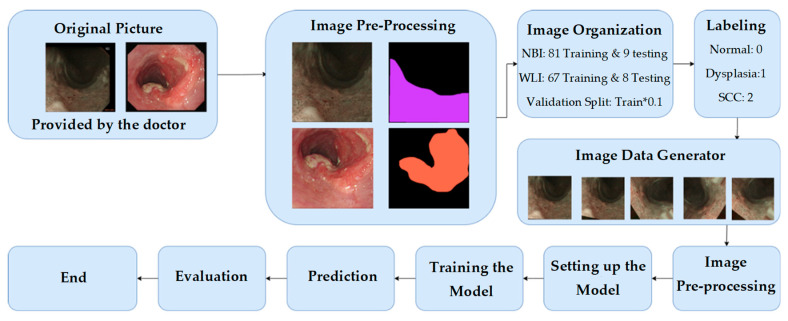
Overall experimental flow chart.

**Figure 2 jpm-12-01204-f002:**
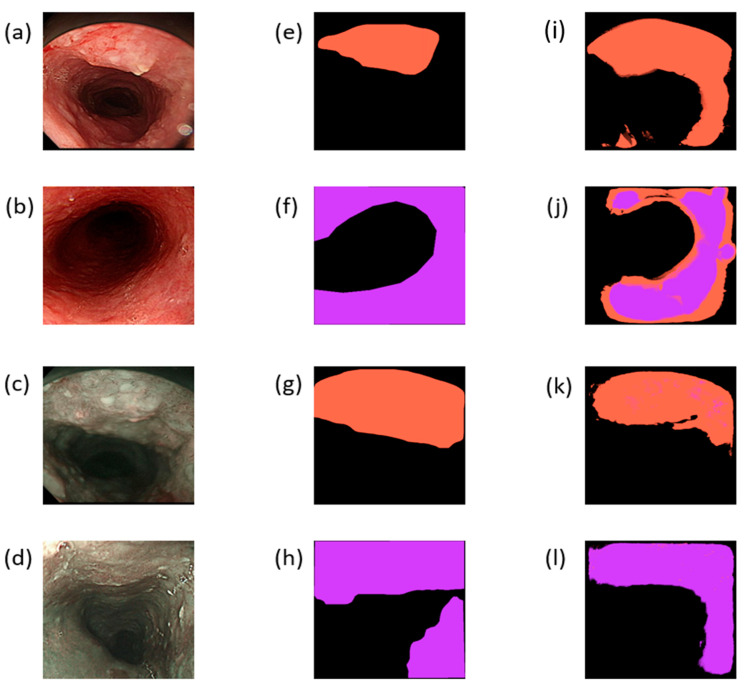
(**a**,**b**) Input images after cropping the original WLI images; (**c**,**d**) input images after cropping the original NBI images. (**e**–**h**) Ground-truth maps marked by doctors based on the corresponding input images. (**i**–**l**) Prediction results of the corresponding graphs; the orange mark corresponds to SCC, whereas the purple mark corresponds to dysplasia.

**Table 1 jpm-12-01204-t001:** Confusion matrix for narrow-band imaging.

**Predicted**		**Normal**	**Dysplasia and SCC**	**Precision**	**F1 Score**	**IoU**
	Normal	525,422	100,490	83.95%	0.857922	67.89%
	Dysplasia and SCC	73,537	439,809	85.67%	0.834833	71.35%
	True Positive Rate	87.72%	81.40%			

**Table 2 jpm-12-01204-t002:** Confusion matrix for white-light imaging.

**Predicted**		**Normal**	**Dysplasia and SCC**	**Precision**	**F1 Score**	**IoU**
	Normal	504,168	82,798	85.89%	0.852622	71.79%
	Dysplasia and SCC	91,495	310,531	77.24%	0.780861	54.48%
	True Positive Rate	84.64%	78.95%			

## Data Availability

Data sharing is not applicable.

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
