# Peer review of "Identification of Early Esophageal Cancer by Semantic Segmentation"

_jpm, 2022, doi:10.3390/jpm12081204_

Round 1
Reviewer 1 Report
The paragraph 2.1.2 Network Architecture must be extended due to its great importance. The readers should be able to understand enterely the applied procedure, for possible comparison or new applications and improvement

Reviewer 2 Report
In this paper, Fang YJ et al has demonstrated how to make the better prediction with accuracy of esophageal squamous cell carcinoma by using the concept of semantic segmentation and U-Net as the basic artificial neural network. Recently, With the help of deep learning architectures like U-Net and CANet, scientist can achieve high-quality results on computer vision datasets to perform complex tasks. Here authors used several narrowband and white light imaging of endoscopic images for Computer-aided diagnosis (CAD). Although authors did not mention about the source of the images. If these are the publicly available data, authors should include the link in the manuscript as the data source. Authors should be more careful about the languages and construction of the sentences. After addressing these points, the manuscript can be considered for the publication.
Page 1, Line 43. What does it mean of the following sentence? -“At present, doctors can-not deduct from the endoscopy of EC”
Authors mentioned, “The precision of dysplasia and SCC categories was 85.67%, which was higher than 174 the accuracy of the normal category” Are these percentages generated from the several images of each category? How many images were included in each group?
What are the inclusion and exclusion criteria for choosing images?
